# Market-based approach to promoting home fortification of diets with micronutrient powder through volunteer frontline health workers: A qualitative study

**Mahfuzur Rahman** [1]*, **Md. Fakhar Uddin**[1], **Tahmeed Ahmed**[1], **Haribondhu Sarma**[2]

**1** Nutrition and Clinical Services Division, icddr,b, Mohakhali, Dhaka, Bangladesh, **2** National Centre for Epidemiology and Population Health, The Australian National University, Canberra, ACT, Australia

* mahfuzur.rahman@icddrb.org

**Data Availability Statement:** Data generated from icddr,b's research can be provided to interested researchers (Recipients) for secondary data analyses upon approval of a Data Licensing

## Abstract

### Background

The potential of market-based approach in distributing micronutrient powder (MNP) through volunteer frontline health workers has been recognized. BRAC, the largest non-government organization (NGO) in Bangladesh, uses Shasthya Shebikas as volunteer frontline health workers to sell MNP for promotion of home fortification (HF) of diets for under-5 children. We aimed to understand the opportunities and challenges of BRAC's market-based approach in promoting HF with MNP.

### Methods

We conducted a descriptive qualitative study in the four selected districts of Bangladesh: Faridpur, Gaibandha, Rangpur, and Rajbari. In-depth interviews, key informant interviews, and focus group discussions were deployed to collect data from purposively-selected Shasthya Shebikas and their immediate supervisors at the field level-Shasthya Kormis, Field Organizers, Managers, and mothers or caregivers of under-5 children. We performed thematic analysis to analyze data.

### Results

We have found that the Shasthya Shebikas play a critical role in promoting access of MNP by the mother/caregivers of children aged 6–59 months at the community level. They counsel the caregivers to seek primary advice about the product and also informally identify undernourished children so that they can receive special attention regarding the use of MNP. However, low profit margins, over-due payments for the sold sachets, poor collaboration with and free distribution of MNP by other NGOs, and inadequate training of Shasthya Shebikas on marketing of MNP have posed major challenges for them to perform as effective sales agents of the product.

Application & Agreement by the icddr,b Data Centre Committee. Interested personnel is recommended to consult this with icddr,b IRB Coordinator Mr. M A Salam Khan (salamk@icddrb. org).

**Funding:** The study was funded by the Children's Investment Fund Foundation (CIFF), UK (Grant number GR-01136). The views, opinions, assumptions, or any other information set out in this article are solely those of the authors and should not be attributed to CIFF or any person connected with CIFF.

**Competing interests:** The authors have declared that no competing interests exist.

## Conclusion

The market-based approach in promoting HF with MNP through frontline volunteer health workers shows much potential, with ample opportunities and few possible challenges. Considering the dynamics, the intervention should fine-tune the factors crucial to maximizing the potentials of Shasthya Shebikas for marketing MNP and promoting HF in order to improve nutrition status of the infants and young children.

## Introduction

The efficacy and effectiveness of micronutrient powder (MNP) in reducing childhood anemia have already been established, although the adherence to the usage of MNP among infants and young children is yet to be consistent [1]. It could be due to the delivery channel in which the interventions on home-fortification with MNP are given. Different countries in the world used different approaches and channels for promoting MNP, and all the approaches have shown potential [2]. Among the delivery platforms to promote MNP at the community level, the market-based approach is one, and it is characterized by sales of product with subsidized price and changing behavior of the consumers [2]. In general, this delivery channel follows social marketing approach where for-profit and not-for- profit strategies are interlinked, although the social benefit is seen in a broad sense [3]. Since one of the salient features of this market-based approach is behavior change communication, the role of volunteer frontline health workers (FHWs) is recognized, particularly in the low-income countries where the programs or interventions are designed to address the problems of maternal and child health [4]. Theoretically, with the involvement of FHWs and with their repeated reinforcement, behavior of the consumers can be modified, and a product can be promoted [5, 6]. We conceptualized such behavioral factors in the market-based approach in promoting MNP with the involvement of volunteer FHWs in "Fig 1". In this approach, there is a potential to reach the underserved and low-income customers, supply MNP at affordable price and create opportunity for the agents (FHWs) to improve their incomes [7].

"Fig 1" shows that volunteer FHWs play a role as sales and behavior change agent in promoting the product like MNP and, as agents, they connect supply side with demand side. So, the interaction between volunteer FHWs and consumers or beneficiaries is the main focus (inside the boxes with darker borderlines). A volunteer FHW counsels by visiting door-to-door for changing the behavior of beneficiaries and promoting the product. However, the reinforcement for making awareness about the benefit of MNP among beneficiaries and promoting it is not straight forward. Counseling by volunteer FHWs is required several times ($c_1$, $c_2$, $c_3$, $c_4$...$c_n$ indicating counseling opportunities in the framework) and, thus, the beneficiaries are likely to be aware of the product, to purchase and trial the product, and to comply with it eventually. Although the interaction between the volunteer FHWs and consumers or beneficiaries is the focus in such a market-based approach, supply of MNP in a subsidized price, training the FHWs, monitoring the activities, and providing incentives on sales from the supply side or program strengthen the capacity of volunteer FHWs and, thus, motivate them for more sales and counseling. On the other hand, the contextual issues in the community can affect the promotion of MNP through volunteer FHWs in the market-based approach.

Studies reveal the potential of market-based approach to promoting home fortification of diets with MNP, with the involvement of volunteer FHWs in the context of low-income countries [8, 9]. A study also shows that marketing micronutrient powder through volunteer FHWs is likely to extend more coverage of MNP compared to other delivery approaches [9]. Studies

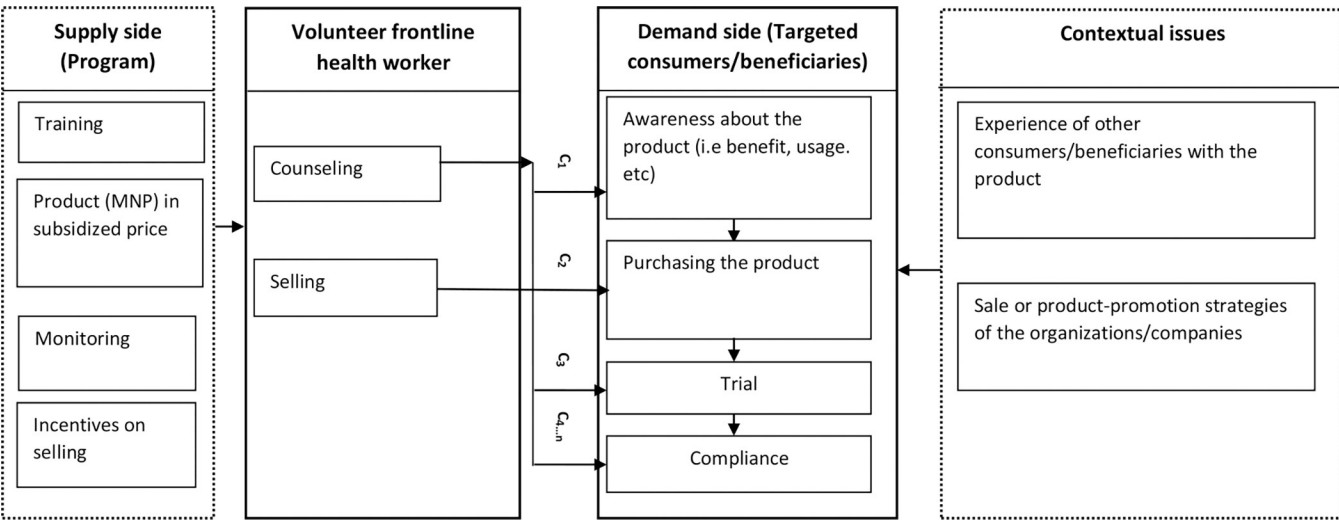

**Fig 1. Conceptualizing market-based approach to promoting a product (MNP) with the involvement of volunteer frontline health worker.**

done in Kenya, Niger, and Nepal even explored some apparent potentials and challenges of market-based approach in promoting MNP [9–11]. A recent study in Bangladesh articulated how volunteer FHWs faced barriers in promoting MNP [12]. However, studies done so far could not explore the underlying factors that can result in challenges and opportunities in a market-based approach to promoting MNP through the volunteer FHWs.

In Bangladesh, different non-government organizations, along with the Government, have taken initiatives to reduce anemia resulting from nutrition deficiency among the pre-school children, in whom anemia prevalence is more than 50% [13]. In partnership with Global Alliance for Improved Nutrition (GAIN) and Renata (a pharmaceutical company), BRAC (the largest NGO in the world, formerly known as the Bangladesh Rural Advancement Committee) has implemented the Maternal, Infant, and Young Child Nutrition (MIYCN) Phase II program in Bangladesh; they have adopted market-based approach to promoting micronutrient powder for reducing anemia among the children aged 6–59 months. The MIYCN Phase II program implemented by BRAC from 2014 to 2018, targeted to cover 15 million children of 6–59 months and aimed to reduce anemia prevalence by 10% from the baseline by home-fortification with micronutrient powder [14]. With the technical assistance of GAIN, Renata is manufacturing Pushtikona-5 (locally-branded name of micronutrient powder) and a single 1 gram sachet of it contains five essential ingredients of micronutrients, such as iron, vitamin A, vitamin C, folic acid, and zinc. Unlike other typical pharmaceutical companies in Bangladesh, BRAC procures Pushtikona-5 from Renata and leverages the existing network of their Shasthya Shebikas (female volunteer frontline health workers) to promote Pushtikona-5 among the mothers or caregivers of children aged 6–59 months. BRAC's Shasthya Shebikas (SSs) are recruited from the targeted communities and provided a three-week basic training on essential healthcare services including promotion of infant and young child feeding practices. They also receive refresher training once in a month. The SSs serve as the first point of contact between communities and BRAC's health and nutrition services. Each SS covers 150–450 households, and an SS visits 14 households per day on average [15]. They receive a small loan to establish revolving funds, which they use to purchase health products such as oral contraceptives, iodized salt, condoms, oral rehydration saline, and so on from BRAC at a small mark-up and sell at a competitive price [16]. The MIYCN Phase II program has been

integrated with the mainstream health program of BRAC and it has reinforced home-fortification of diets with Pushtokona-5 among the children of 6–59 months. In addition to selling Pushtikona-5, the SS provided caregivers with nutrition education and counseling to ensure their compliance with HF while using Pushtikona-5 [12].

BRAC procures a sachet of Pushtikona-5 from Renata at a cost of 1.80 BDT (USD 0.022) and sells it to their SSs at 1.85 BDT (USD 0.023). In doing so, BRAC does not make any profit but charges the transportation cost they require to send the Pushtikona-5 sachets at the community level from the depot of Renata. The SSs sell a sachet of Pushtikona-5 at a price of 2.5 BDT (USD 0.031) to the mothers or caregivers of children and, thus, make a profit of BDT 0.65 (USD 0.008) from each sachet of Pushtikona-5. Although the SSs are the volunteers as they are not paid for their work, they make profit from selling Pushtikona-5 and also receive incentives after achieving their sales targets. An SS receives BDT 30 (USD 0.374) if she can ensure the feeding of 30 sachets for a child. Thus, she receives BDT 120 (USD 1.498) if she can ensure the feeding of 120 sachets for a child within one year. At the community level, the overall activities of the SSs are monitored by their immediate supervisors-Shasthya Kormi (paid frontline health worker of BRAC) and Field Organizers (paid staff of BRAC).

Despite having financial and non-financial incentives (i.e social recognition), the SSs get de-motivated due to programmatic and contextual factors that pose challenges. Although different studies highlighted the overall challenges and facilitators of volunteer FHWs involved in different programs on overall maternal and child health [4], the challenges and facilitators that frontline volunteer health workers face during sale and promotion of micronutrient powder in the community was not rigorously and exclusively looked into. Studies done so far have indicated programmatic and contextual factors relating to the performance of the FHWs [14], how frequent home-visits by the FHWs can improve the coverage of MNP [17] and how incentives can play role in shifting focus to promoting a particular product [18]. However, none of the studies have explored the underlying factors that can result in opportunities and challenges of the market-based approach to promoting home-fortification with MNP through the FHWs. So, the present study aimed to explore the opportunities and challenges of the market-based approach to promoting MNP through frontline volunteer health workers; this was largely an unaddressed issue that had so far been overlooked. This study was a part of the large evaluation of maternal, infant and young child home fortification program in Bangladesh, and we aimed at exploring the opportunities and challenges to inform such programs in course-correction. This paper describes the opportunities and challenges of this approach so that it can help in minimizing the gaps in effective promotion of home fortification of diets with micronutrient powder through volunteer FHWs in Bangladesh and in other similar settings as well.

## Materials and methods

### Study design and setting

This study was part of a large concurrent evaluation of the MIYCN home fortification program in Bangladesh. A detail description of the concurrent evaluation strategy has been reported elsewhere [19]. For conducting the study, we applied a qualitative approach. Although the MIYCN Phase II program was implemented in 34 districts in Bangladesh for the promotion of Pushtikona-5 in order to reduce iron-deficiency anemia among children aged 6–59 months, we did the study in four districts: Faridpur, Gaibandha, Rangpur, and Rajbari from September to December 2015.We selected these four districts because two of these (Faridpur and Gaibandha) were low-performing districts and the other two (Rangpur and Rajbari) were high-performing ones in terms of both reducing anemia among children aged 6–59 months and promoting Pushtikona-5, with FHWs visiting households. In Rangpur and Rajbari districts,

anemia reduced by 29. 31 and 28. 17 percentage points from baseline to endline, respectively. On the other hand, in Faridpur and Gaibandha districts, anemia reduced by 12.35 and 19.75 percentage points from baseline to endline, respectively. We collected information on performance of the areas from a periodic assessment of BRAC's MIYCN home fortification program done by icddr,b (International Centre for Diarrheal Disease Research, Bangladesh) in the implementation areas. We set some criteria to understand the performance of BRAC's program, such as coverage of Pushtikona-5 in the areas, i.e percentage of mothers or caregivers who had heard of Pushtikona-5, who had ever fed it to their children, and who had fed at least one sachet of Pushtikona-5 in the last one week and percentage of households in the areas that received a visit ever or a visit during the last year by the volunteer frontline health workers of BRAC. Where percentages of the abovementioned indicators were low, we considered the areas to be low-performing and where the said percentages were high, we considered the areas to be high-performing.

## Sampling

We purposively selected mothers of children aged 6–59 months, Shasthya Shebikas (SSs), and Shasthya Kormis who were the immediate field-level supervisors of SSs as well as other BRAC staff, like Field Organizers, Upazila (Subdistrict) Managers and District Managers. We included the mothers who ever received or partially received or never received advice or services on Pushtikona-5 from SSs; included the FHWs (SSs), their immediate field-level supervisors- Shasthya Kormis, and other BRAC staff, like Field Organizers, based on their duration of job experience, education, and age in order to maximize the variation in data. Based on literature and previous work experiences we planned to conduct a total of 8 in-depth interviews (IDIs) with the caregivers of the children, 4 IDIs with SSs, 4 IDIs with Shasthya Kormis and 4 IDIs with Field Organizers. In addition, we planned to conduct a total of 4 FGDs with the Field Organizers and 4 FGDs with the Shasthya Kormis. However, we had to conduct a total of 15 IDIs with the caregivers of the children to reach the data saturation point. The number of IDIs with the SSs, Shasthya Kormis and Field Organizers, and FGDs with the Shasthya Kormis and Field Organizers we planned were found to be adequate for data saturation from provider perspective.

In BRAC's program design, there was a manager in each district and each upazila under that district. We selected two District Managers and two Upazila Managers from high-performing areas and, similarly, two District Managers and Upazila Managers from the low-performing areas. Thus, we selected a total of 4 District Managers and 4 Upazila Managers as key informants from the study areas. We interviewed them as key informants to get an overall idea from programmatic perspective of the market-based approach to promoting Pushtikona-5 through SSs. "Table 1" shows the number of IDIs, KIIs and FGDs conducted among the respective participants.

**Table 1. Number of IDIs, KIIs and FGDs conducted among the respective participants.**

| Data collection methods | Total number | Respective participants (number) |
|---|---|---|
| In-depth interview | 27 | Caregivers of the children (15), Shasthya Shebikas (4), Shasthya Kormis (4), Field Organizers (4) |
| Key informant interview | 8 | District Managers (4), Upazila Managers (4) |
| Focus group discussion | 8 | Field Organizers (4), Shasthya Kormis (4) |

## Data collection

In this study, we intended to collect data on the experiences of the study participants relating to the challenges and barriers of the market-based approach to promoting micronutrient powder through volunteer FHWs. A team consisting of six members was involved in developing separate guidelines for conducting face-to-face interviews, focus group discussions, and key informant interviews with the study participants. All of them had postgraduate degrees in anthropology. The first author (MR) of this paper generated the research idea, oversaw the data collection process, analyzed data, and prepared the manuscript. Two authors (MU and MR) experienced in qualitative research were fully involved, starting from developing the guideline to the coding of data. One of the authors (HS) who was the principal investigator of the study and the senior author (TA) provided overall guidance for conducting the study, drafting and reviewing this paper.

In-depth interviews were the main research technique used in collecting qualitative data. We also conducted focus group discussions (FGDs) and key informant interviews (KIIs) with BRAC's program staff and managers respectively. While developing the guidelines for conducting interviews and focus group discussions with program staff, such as Managers, Field Organizers, and Shasthya Kormis, we did it by focusing on programmatic aspects, such as training, monitoring, MNP supply, etc. The guidelines for conducting interviews with volunteer FHWs and beneficiaries, i.e mothers or caregivers of children, were developed by focusing on contextual aspects and their lived experiences. However, the guidelines were flexible in incorporating questions, if required, based on the findings from interviews, even during data collection at the study sites.

Average time for an interview was 46 minutes. Average number of participants in the FGDs was 7, ranging from 6 to 8. In order to avoid any disruptions, data collection was done at the convenient locations of the participants, such as residences or workplaces.

## Data analysis

Interviews were recorded and transcribed in Bangla language immediately after data collection. The transcripts of the interviews were read thoroughly for familiarity with the data and for identifying the issues that needed to be explored. Furthermore, the interviewers sent field-impression notes or memos to the investigator in order to get feedback on what issues they needed to investigate more in-depth. The investigator provided feedback on those immediately so that the interviewers could go more in-depth on those issues from the next interviews. In this iterative way, initial analysis began during data collection. Then the team members sat together to identify the deductive codes generated from research objectives and coded on the textual format of data in the transcript individually. They exchanged the coded transcripts with one another and checked if the coding was consistent. We used Microsoft Word Document to tabulate the textual data and manually codded on those.

After completion of data coding manually, the common patterns were identified, and data were analyzed thematically. Findings were triangulated with data collected from different sources (respondents) in order to check the validity of data. To report FGD and interview data, we followed Consolidated criteria for Reporting Qualitative research (COREQ) checklist [20] ("S1 Checklist").

## Ethics statement

Our study protocol was reviewed and approved by the Institutional Review Board of icddr,b which is comprises of Research Review Committee and Ethical Review Committee. We obtained written informed consent from all participants before conducting interviews or

discussions with them. Before taking consent from the participants, the objectives of the study, the risks and benefits of their participation, and their rights of withdrawal or not responding to any questions if they would feel embarrassed were clearly explained.

## Results

There were a total of 15 mothers or caregivers with whom we had in-depth interviews; their average age was 22.84 years, ranging from 18 to 33 years. All of them were housewives. Most of them had more than 5 years of education.

The volunteer FHWs who participated in this study had an average age of about 45 years, and most of them (3 out of 4) had been working with BRAC as FHWs for more than five years. We also found an FHW who had been working with BRAC for more than thirteen years. We found two FHWs who had no formal education.

Most of the Shasthya Kormis had completed 10 years of education and worked for BRAC for more than 5 years. We found only one Shasthya Kormi who had joined BRAC one year ago. Most of the Field Organizers, Upazila (Subdistrict) Managers, and District Managers who participated in this study had obtained postgraduate degrees. However, most of the Field Organizers had joined BRAC about two years prior to the day of interview with them whereas most of the Upazila Managers and District Managers had joined BRAC about 8 years ago.

Although the main evaluation was conducted in the low and high performing districts, in this paper we focused on overall market-based approach to promoting home fortification of diets with MNP through FHWs regardless of study areas. In addition, during data analysis we did not find any differences in the promotion of home fortification of diets with micronutrient powder through FHWs in between high and low performing districts. Therefore, we present the findings below thematically that reflect market-based approach to promoting MNP.

The SSs-the volunteer FHWs in BRAC's program-play the role as sales agents in the market-based approach to promoting MNP and in the interaction between supply side and demand side. Such an interaction and this market-based approach can be affected by some other contextual issues. So, the results of this study presented in this paper encompasses supply side, demand side, and contextual issues relating to the opportunities and challenges of this approach, with the involvement of FHWs in promoting MNP.

### Opportunities of market-based approach to promoting micronutrient powder through volunteer FHWs

This study explored some of the opportunities that a market-based approach to promoting MNP had when an FHW would play the role as a sales agent. Mothers' accessibility to a new product, like MNP; the way to seeking primary advice about MNP; and informal identification of undernourished children by the FHWs in the community came up as the opportunities of market-based approach to promoting HF with micronutrient powder through volunteer FHWs in Bangladesh.

### Accessibility of mothers to Pushtikona-5

Mothers of children aged 6–59 months, who received visit of the SSs, had easy access to Pushtikona-5, as it was available with the SSs who lived in their communities. Mothers of children who ever received Pushtikona-5 from SSs perceived that they could easily get Pushtikona-5 because the SSs made door-to-door visits within the community and sold Pushtikona-5 they required. However, some of the mothers also mentioned that they obtained Pushtikona-5 from the homes of SSs. A mother said:

*She (SS) lives in our village. I knew her. After feeding one box of Pushtikona, I ran out. Then I went to her (SS) home and obtained some sachets of Pushtikona from her. After a few days, she visited my household and gave me more sachets of Pushtikona.*

We also found mothers in the community, who conveyed message to the SSs that they had run out of Pushtikona-5. The SSs would then visit those households and sell Pushtikona-5 to them.

Ensuring availability of Pushtikona-5 in the households of SSs would ensure the accessibility of Pushtikona-5 in the community and, thus, it would promote sales of Pushtikona-5 as mentioned by one of the District Managers in an interview. The District Manager said:

*During the meeting, I told the SSs to keep the sachets of Pushtikona in their households. I told them that selling Pushtikona is not the first concern; rather they should ensure the availability of Pushtikona in their households. We found that, when Pushtikona was available in the homes of SSs, the sachets were sold sooner or later. . .how can we sell if we do not have the product?*

To ensure the availability of Pushtikona-5 at the homes of SSs, some extra boxes were kept in the homes of Shasthya Kormis who could supply these to the SSs immediately if there was any shortage.

During the interview, an SS claimed that she could always provide Pushtikona-5 when a mother wanted it from her; she said:

*Whenever a mother wants (Pushtikona), I immediately give her. Even if I do not have (it) in my house, I go to Shasthya Kormis and bring Pushtikona (for that mother).*

The study also revealed that this approach to promoting MNP through FHWs had opportunity to ensure the accessibility to the product among the mothers residing in hard-to-reach areas. A Field Organizer, in an interview, mentioned:

*In the hard-to-reach areas where there is a lack of formal healthcare provider, they (SSs) are working more and selling more general drugs and Pushtikona too, even in absence of our supervision because we cannot always visit there.*

## Way to seeking primary advice about MNP from the FHWs

Mothers of children who ever received Pushtikona-5 or heard of Pushtikona-5 mentioned that SSs were the sources from whom they received it or heard about it. Mothers mentioned that they received advice regarding Pushtikona-5 and its efficacy in reducing anemia, increasing the appetite of children, and its ability to make the children meritorious.

Two mothers also mentioned that they sought advice from the SSs as they were from their own communities and, so, they were trustworthy. One of them said:

*She (SS) is from our community. She will not do anything harmful for my child. Our children are their children too.*

We also found that the SSs, who could not read, suggested mothers or siblings of the children to read the brochure kept in the packet of Pushtikona-5 in order to get clarification on how to use it and what the benefits of using it are. Upon being asked what happened if the

mothers also could not read, one of the SSs added that she requested the neighbors to read out the information so that the mothers could understand it.

One of the SSs claimed that receiving advice from the SSs also helped mothers feel like they were more in their comfort zone compared to receiving advice from medical professionals.

## Informal identification of undernourished children

The study suggested that, although the SSs were not formal healthcare providers, they could identify the undernourished children without applying any formal diagnostic procedure (such as any formal check-up), which the formal healthcare providers would generally apply. We found SSs who could identify the undernourished children in their community by looking at the physical appearance of the children and discussing informally with the mothers about the problems of their children. The SSs were found to be intended to identify undernourished children and promote Pushtikona-5 to them and, as lay health workers, they could do that because they were BRAC's SS or *Dhai* (unskilled birth attendant) or population-related information-collector in the community. Although the SS did not have any training on identifying mal-nourished children, they could do that observing pale face and eyes, rashes on the skin, distended abdomen, and thin body of the children. One of the SSs said:

> *When I notice a child with very pale face and eyes, I ask the mother if her child suffers from diseases, like diarrhea. Then the mother immediately asks me how I've understood that. I tell her that her child looks undernourished. I also tell her that I have Pushtikona that contains zinc. Have you ever seen the advertisement of zinc tablet in TV, which is helpful for diarrheal patients? Thus, I adopt these tactics, and the mother takes the Pushtikona from me.*

Some of the SSs also added that, when the mothers reported about frequent suffering of their children from diseases, like diarrhea, fever, or having loss of appetite, they also considered those children to be undernourished and promoted Pushtikona-5 especially among them.

## More sales of and compliance with MNP through the provision of incentives

Study revealed that, if incentives are provided in time on the basis of sales and feeding a certain number of sachets to a child, this market-based approach would have the potential to increase the sales and compliance with MNP as well.

A Field Organizer, in an interview, said:

> *They (BRAC's paid staff) are providing incentives, and the SSs are being motivated after receiving incentives. When an SS receives incentives, she works more (on selling, counseling, etc.).*

This study also explored some challenges of the market-based approach to promoting MNP through volunteer FHWs, such as low profit margins and over-due payment for the sold sachets of Pushtikona-5, that demotivated the SSs to sell it. Moreover, distribution of other MNPs by other organizations at lower price or even free of charge in the working areas of SS created further challenge. Delay in disbursement of incentives on sales of Pushtikona-5 demotivated the SS, which came up as another challenge.

## Low profit margins

When we asked the SSs about what health commodities they sold, none of them mentioned Pushtikona-5 at first. They mentioned other general drugs for fever, dysentery, and diarrhea,

and other vitamin products in particular. They also added that they could make more profit by selling vitamins and other drugs compared to Pushtikona-5 and so they preferred selling those drugs. One of the SSs mentioned:

> I prefer selling vitamins and drugs for fever because I can earn more from those.

Study revealed that the SSs could not sell Pushtikona-5 at high prices as they could in the case of other drugs or vitamins. The price was clearly written on the packet of Pushtikona-5 and so they could not sell those at a high price, most of the SSs mentioned. Rather they had to sell them at lower price as they had sales targets for Pushtikona-5, such as two or more boxes in a month to receive the incentives.

### Over-due payments for sold sachets of Pushtikona-5

The study revealed that one of the major challenges for the SSs was over-due payments (pay later) for the sold sachets of Pushtikona-5. This resulted in the scarcity of their revolving fund, with which they would purchase other essential health commodities, including Pushtikona-5 from BRAC. We explored that, although the SSs were volunteers, all of them were motivated by their immediate supervisors- Shasthya Kormis and Field Organizer to sell more Pushtikona-5. They became target-oriented on selling Pushtikona-5 so that they could make more profit. For doing so, the SSs, sometimes, sold Pushtikona-5 in due time to promote it; sometimes, they failed to collect the due payments.

However, we found one case where the SS and Shasthya Kormi worked together for their mutual interest. The Shasthya Kormi helped the SS collect the over-due payment, and the SS helped her identify the pregnant women in the community.

This study also revealed that, sometimes, the beneficiaries availed the opportunity to buy Pushtikona-5 from SSs keeping the payment due, rationalizing that the SSs and the beneficiaries lived in the same communities and they could pay the dues when money will be available in their hands. One of the SSs mentioned:

> A mother of a child received Pushtikona from me on due payment but, later on, the father of the child did not pay, arguing that they were supposed to get Pushtikona free of cost as Pushtikona was generally supplied by Government.

This study also explored why the community people perceived that any product, like MNP, was to be supplied free of charge. The reasons are explained below.

### Competing with other organizations that sell MNP at lower price or free of charge

Due to lack of collaboration, other NGOs were also distributing MNP but free of charge in the same areas where BRAC's program was being implemented. Thus, the beneficiaries from the communities perceived that Pushtikona-5 was a product like other MNPs that should also be distributed free of charge. In an FGD, a Shasthya Kormis mentioned:

> In our areas, an NGO, named X, is distributing MNP free of cost. They are not distributing among all the mothers of children under-five years of age. Suppose, they distribute to two mothers out of ten. When our SSs visited the remaining eight mothers, they complained that the NGO X was distributing MNP free of cost, so why they would buy it from BRAC.

This study also revealed that a social marketing company is selling MNP at a lower price through market outlets; this is also creating challenges for the SSs in selling Pushtikona-5 at a comparatively higher price. An Upazila (Sub-district) Manager, in an interview, said:

*While a social marketing company is selling a box of MNP at BDT 30 (USD 0.374) through market outlets, we are selling our one box of Pushtikona at a cost of BDT 75 at the community level. When people ask us why we are selling at a higher cost, we cannot respond.*

## Less capability of Shasthya Shebikas and other frontline health workers on marketing MNP

We explored that the training SSs received on promoting Pushtikona-5 was not adequate for marketing it in the community. Most of the SSs acknowledged that they were not skilled enough to motivate the mothers or other family members, like grandmothers or fathers of the children when they inquired more about the product. We even found an SS who was afraid of further convincing the mothers when the children had side-effects, like dark stool or constipation after consuming Pushtikona-5. However, she mentioned that she did not feel fear when she sold Pushtikona-5 among her relatives.

She also added that when she was unable to convince the mothers about harmlessness of the side-effects of Pushtikona-5, the mothers spread rumors in the community that made the other mothers reluctant to purchase Pushtikona-5 for their children.

The SS said:

*We still cannot make the mothers understand well about Pushtikona. If we had the opportunity to receive more training on this, we would make them understand better.*

Other BRAC staff (Shasthya Kormis and Field Organizers) who supervised the activities of SSs also acknowledged that they themselves could not say anything when the mothers claimed that their children received good feeding; so, how could the SSs convince those mothers. This finding coincided with the findings we had from an interview with a mother who did not receive any message about Pushtikona-5. She said:

*They (SSs) did not tell me anything about it (Pushtikona-5). Perhaps, they thought that my child ate good foods and, so, he would not need it.*

The study found that the mothers who received Pushtikona-5 from SSs informed the SSs when their children were unwillingly to take it. However, most of the SSs could not convince the mothers further or demonstrate the process that the mothers could try to feed Pushtikona-5 persistently to their children. A mother mentioned:

*My child did not want to eat this. So I did not feed him further...I informed her (SS) about this. She replied that I should try.*

## Delay in disbursement of incentives on Pushtikona-5 sales demotivates the FHWs

Study revealed that supply side or programmatic issue itself created challenge to promoting Pushtikona-5 through volunteer FHWs. We found that the SSs were supposed to be visited directly by Shasthya Kormis for monitoring purpose at least once a month, and the SSs received random visits by Field Organizers in a similar way. However, the study explored that

the visits by Shasthya Kormis and Field Organizers varied and were irregular due to the unequal distribution of SSs in the working areas of Shasthya Kormis and Field Organizers. So, it revealed that, although there was a provision of incentives on selling Pushtikona-5, there were delays in receiving incentives as they were given after monitoring and confirmation by BRAC staff (Shasthya Kormis and Field Organizers), which was not done on a regular basis. Inadequate number of Field Organizers, who were basically involved in monitoring, came up as one of the major reasons for delay in monitoring. Thus, SSs did not get incentives on time and became demotivated to sell Pushtikona-5.

A Field Organizer mentioned:

*Due to our workload, we cannot monitor and without monitoring we cannot provide incentives. We have such a case (where an SS completed target for incentives two months ago) where we had to provide incentives to the SS after two months.*

## Discussion

This study has explored some of the opportunities and challenges of market-based approach to promoting home fortification of diets with micronutrient powder through volunteer frontline health workers, which can all be taken into account for making this type of approach more effective as it has already been recognized as a successful approach. This market-based approach has not only potential to be sustainable [2] but also to be cost effective [21].

Although the results of this study coincide with the findings of other studies that have already identified some opportunities and challenges from the perspectives of individual, societal and institutional level [14], this study has explored the supply-side, demand-side and contextual issues around the market-based approach. One of the major opportunities of the approach that the study illustrates is the accessibility to micronutrient powder among the mothers or caregivers of 6–59 months old children. A study in Nepal shows that coverage of micronutrient powder is comparatively higher when it is channeled through community volunteers compared to the health facilities [9]. Accessibility may lead to increased coverage of micronutrient powders as the studies show that the mother or caregivers are not required to travel to get the MNP from the volunteer frontline health worker, she being a member of her own community [9]. A study done in Bangladesh has also shown that the coverage of MNP is associated with the home-visits by volunteer FHWs [17]. In addition, FHWs can be the essential sources of MNP since outlets or pharmacies are rare in the resource-poor rural settings [11].

Compared to the outlets or pharmacies, the SSs provide more comfort to the mothers or caregivers when receiving advice regarding the use of micronutrient powder, which also corresponds to the findings of a study [4]. Accessibility to such a commodity or service not only expands the coverage of the commodity but also, in turn, may reduce the workload of the health professionals [4]. When the mothers or caregivers of children go to the SSs for receiving services regarding MNP as a preventive measure from anemia, they may not need to visit other formal healthcare providers. Thus, it may reduce the workload of the formal health professionals.

The workload of the formal health professionals can also be reduced if there is an opportunity for informal identification of undernourished children by the SSs in the community. Superficial identification of undernourished children through observation of their physical appearance and then promoting MNP to them can be a strategy for preventive measure not only against the nutritional anemia but also against severe malnutrition that requires management from formal health professionals.

Despite these opportunities, this study reveals some challenges which need to be addressed for the promotion of micronutrient powder through market-based approach where volunteer frontline health workers are involved. The challenges faced by the volunteer FHWs are rooted in programme, organizational and societal level [14]. The challenges that the study shows correspond to some of the findings from the synthesis of qualitative evidence of a study which, in general, represents the challenges of lay health workers involved in the overall maternal and child health programs [4]. One of the challenges that coincide with the findings from that review is the reluctance of the health workers to provide health services from the fear of being unsuccessful [4]. However, our study shows where this fear of being unsuccessful is rooted and how it does vary in different contexts. In the context of this study, we found that the fear of being unsuccessful in achieving an outcome from promoting home fortification with micronutrient powder stems from their lack of training on strategies for convincing the mothers or caregivers who complain about some general side-effects, like dark stool. Their fear of being blamed is less when they sell Pushtikona-5 among their relatives, which indicates that the comfort level in interactions with the mothers or caregivers of children may be an underlying factor in promoting micronutrient powder through the market-based approach. However, special training may be required to ensure more depth of interactions between volunteer FHWs and mothers or caregivers [14]. The challenges discussed above can be addressed with some programmatic efforts. These efforts are also warranted in sensitizing the community as the study reveals over-due payment on the sold Pushtikona-5 as a challenge that may demotivate the volunteer FHWs from promoting the product.

Motivation of the FHWs also depends on the incentives they receive from sales of MNP and the profit they make [14]. A study has indicated that incentives on the sales of MNP motivates the FHWs to make more home-visits and thus to increase the coverage of MNP [17]. The findings of this study point out that low profit margin on selling Pushtikona-5 and delays in receiving incentives on sales demotivate the SSs. Programmatic efforts are required via timely monitoring and supervision of the activities related to promoting Pushtikona-5. A study done in Kenya recommends strong monitoring and supervision of the programmatic activities for the sustainability of market-based approach in promoting micronutrient powder [22]. Evidence from this study suggests that monitoring should not be limited to only overseeing the activities of marketing Pushtikona-5 but this can also be a functional process aimed at ensuring sales incentives for the FHWs. Thus, this can motivate them to sell more Pushtikona-5. In addition, coordination with other organizations that are also distributing micronutrient powders in the community can minimize the challenges faced by the volunteer FHWs, as the study shows that free distribution of MNPs by other organizations within the same community places the workers in dispute with the beneficiaries. Therefore, coordination among the implementing partners is warranted to promoting home fortification of diets with MNP in the communities and the government should have control over the price points of it. Since the Government of Bangladesh has already adopted the strategy on MNP program implementation and prioritized the areas of action including coordination between the Ministry of Health and Family Welfare and other implementing organizations [23], evidences generated from this study are likely to contribute to the sustainability of such a program.

This study also recommends to use a comprehensive implementation science strategy [24] that may help the policy makers and implementers get in-depth insights about the challenges and opportunities of market-based approach to promoting home-fortification with MNP through the volunteer FHWs. The flexible framework can show the implementers a pathway on how the facilitators could be strengthened and the challenges can be overcome while scaling up the market-based approach to promoting home-fortification with MNP in real setting.

## Strengths and limitations

This descriptive qualitative study, for the first time, explored the opportunities and challenges of market-based approach to promoting home fortification with MNP through volunteer frontline health workers. The study maximized the variations in selection of the participants from different tiers to triangulate the findings and get a holistic view of the results. However, the study could not capture the contextual issues of the volunteer frontline health workers comprehensively, relating to the challenges they faced and the opportunities they had, due to the short duration of data collection. Ethnographic study is required for an in-depth and holistic understanding of the challenges and opportunities from programmatic and contextual perspectives.

## Conclusion

The potential of market-based approach in promoting HF with MNP through frontline health workers is subjected to a number of opportunities and challenges. The opportunities include selling MNP among the low-income customers in affordable price, creating easy accessibility to the agents (FHWs) and thus to receive the products and counseling on how to use the product. However, overdue payment, delay in disbursement of incentives or irregular incentives, competing market for the layman volunteer FHWs pose challenges for the market-based approach to promoting home-fortification with MNP. Considering such factors, programmatic adjustments are required. Shasthya Shebikas of BRAC can be utilized for marketing MNPs and promoting HF to improve the nutrition status of infants and young children in Bangladesh, however, skill development training and income generating activities for the FHWs are warranted.

## Supporting information

**S1 Checklist. Consolidated criteria for Reporting Qualitative research (COREQ) checklist.** (PDF)

## Acknowledgments

We thank all the participants for participating in this study. We also thank the study team members who were involved in data collection. icddr,b acknowledges with gratitude the commitment of The Children's Investment Fund Foundation (CIFF) to its research efforts. icddr,b is also grateful to the Governments of Bangladesh, Canada, Sweden and the UK for providing core/unrestricted support.

## Author Contributions

**Conceptualization:** Mahfuzur Rahman.

**Data curation:** Mahfuzur Rahman, Md. Fakhar Uddin.

**Formal analysis:** Mahfuzur Rahman, Md. Fakhar Uddin.

**Funding acquisition:** Haribondhu Sarma.

**Investigation:** Mahfuzur Rahman, Haribondhu Sarma.

**Methodology:** Mahfuzur Rahman.

**Project administration:** Mahfuzur Rahman, Haribondhu Sarma.

**Supervision:** Mahfuzur Rahman, Tahmeed Ahmed, Haribondhu Sarma.

**Writing – original draft:** Mahfuzur Rahman.

**Writing – review & editing:** Md. Fakhar Uddin, Tahmeed Ahmed, Haribondhu Sarma.

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
