## [Decision Letter · Decision Letter 0]

3 Oct 2022

PONE-D-22-04029Market-based approach to promoting home fortification of diets with micronutrient powder through volunteer frontline health workers: A qualitative studyPLOS ONE

Dear Dr. Rahman,

Thank you for submitting your manuscript to PLOS ONE. After careful consideration, we feel that it has merit but does not fully meet PLOS ONE’s publication criteria as it currently stands. Therefore, we invite you to submit a revised version of the manuscript that addresses the points raised during the review process.

We look forward to receiving your revised manuscript.

Kind regards,

Ilias Mahmud, PhD

Academic Editor

PLOS ONE

Journal Requirements:

Reviewers' comments:

Reviewer's Responses to Questions

**Comments to the Author**

1. Is the manuscript technically sound, and do the data support the conclusions?

Reviewer #1: Partly

Reviewer #2: Yes

2. Has the statistical analysis been performed appropriately and rigorously? 

Reviewer #1: N/A

Reviewer #2: N/A

3. Have the authors made all data underlying the findings in their manuscript fully available?

Reviewer #1: No

Reviewer #2: Yes

4. Is the manuscript presented in an intelligible fashion and written in standard English?

Reviewer #1: Yes

Reviewer #2: Yes

5. Review Comments to the Author

Reviewer #1: This paper describes a qualitative evaluation of an existing market-based intervention promoting the sale of MNP by FHW in four districts of Bangladesh, with two identified as high performing and two identified as low performing. Overall, the writing and analysis could be tightened and strengthened. There are some interesting and relevant findings, but the design (high and low performers) is not relevant in the way the results are reported and it’s unclear if sample sizes are sufficient for the stratification despite authors reporting they reached saturation. More details are needed regarding the original program and evaluation methods. The conclusions focus on more implementation science research needed but there’s a gap/role for government and oversight of interventions and policies that warrant mentioning. Some specific suggestions and questions below.

It seems the FHW also play a role as trouble shooters when caregivers experience problems with MNP, but not clearly stated in initial descriptions. Also, text doesn’t describe how FHW link IYCF/MNP and how/if support other IYCF so please clarify. Both WHO and HF-TAG state that MNP should be integrated with IYCF programs.

Reader doesn’t know what this means “(c1, c2, c3, c4…cn),” possibly refers to counseling opportunities?

What was the suggested MNP intake regimen being suggested by FHW? Overall, there’s no description of how FHW were trained and how program was supposed to be implemented and be effective, other than high level saying FHW sold MNP to mothers. Reader finds out part way through the paper that FHW sell other products. Authors assume reader understands their program without providing information. More details about program design would help reader understand results (e.g., how are FHW recruited, trained, how often are they supposed to get supplies to sell, interact with caregivers? Authors reported trouble counseling mothers, but were they even trained to address this?)

Switches from FHW to CHW mid-way through paper.

Further details (e.g., #, %) about how high/low performers and low sales were defined and assessed, this would improve reader understanding of differences between the two groups. Also, paper explicitly says anemia reduced as a criteria for high/low but no evidence provided.

Reports number when saturation occurred but not planned number of interviews, what were potential # to be interviewed? It’s unclear in methods total number of FGD vs individual interviews (or other types) conducted, suggest presenting in a table with numbers described for each (e.g., was it 2 FGD with 15 mothers total, for example? Noted later some of this is stated in results, but that’s too late and overall unclear to reader how many individual vs FGD vs other). Noting samples sizes are low and should be stated as a limitation, minimum sample size for stratification in qualitative data collection is 3 (and that’s low) and authors note 2 per multiple stratifications, recognizing limited number of districts. Achieving saturation at 2 interviews is a limitation.

Was a software used for coding and analysis?

Suggest avoiding acronyms for BRAC job descriptions - for readers unfamiliar it’s a challenge to remember what they mean and there are multiple used throughout the paper.

“On due payment” means to pay later?

It seems there’s uncontrolled delivery of interventions in the area that are not coordinated or known (?) by government with so many delivering MNP for free or different price points. It would be impossible to be successful with higher cost product in this environment unless others were perceived inferior. Deserves mention of role of govt to approve interventions and coordinate etc.

The design was focused on high and low performers, which is a good idea, but the authors do not present/discuss results using that stratification. Were there no differences? Either way, this should be noted in results or discussion.

Reference #1 has been updated for WHO in 2020 and 2011 is old/out of date. Appreciate there has not been too much published on market-based sales of MNP, but noted many references are ~10 y old and the authors tend to cite themselves.

Reviewer #2: I found this paper relevant and useful for the health sector of Bangladesh. A particular timeline of intervention and how it impacted the overall adaption of strategies into government level of sustainability could have been added. However, the paper as it is currently publishable and will benefit global research community.

6. PLOS authors have the option to publish the peer review history of their article (what does this mean?). If published, this will include your full peer review and any attached files.

Reviewer #1: No

Reviewer #2: **Yes: **Shamim Ahmed

---

## [Author Response · Author response to Decision Letter 0]

20 Nov 2022

Responses to reviewers’ comments

We appreciate the very thoughtful reviews of the previous version of the manuscript. We have updated the text in response to the reviewers’ queries and feedback. A point-by-point response to each of the reviewers’ comments is included below. We believe these changes have substantially improved the manuscript. We hope you will find this revised manuscript appropriate for publication in PLOS ONE. Many thanks for your consideration.

Reviewer #1: This paper describes a qualitative evaluation of an existing market-based intervention promoting the sale of MNP by FHW in four districts of Bangladesh, with two identified as high performing and two identified as low performing. Overall, the writing and analysis could be tightened and strengthened. There are some interesting and relevant findings, but the design (high and low performers) is not relevant in the way the results are reported and it’s unclear if sample sizes are sufficient for the stratification despite authors reporting they reached saturation. More details are needed regarding the original program and evaluation methods. The conclusions focus on more implementation science research needed but there’s a gap/role for government and oversight of interventions and policies that warrant mentioning. Some specific suggestions and questions below.

It seems the FHW also play a role as trouble shooters when caregivers experience problems with MNP, but not clearly stated in initial descriptions. Also, text doesn’t describe how FHW link IYCF/MNP and how/if support other IYCF so please clarify. Both WHO and HF-TAG state that MNP should be integrated with IYCF programs.

Response: Thank you for your valuable comment. In the revised manuscript, we have described how a FHW under the home fortification of diets with MNP program has been involved in multiple tasks including promotion of IYCF practices and selling health commodities such as oral dehydration saline, iodized salts etc. along with MNP. (Page 7)

Reader doesn’t know what this means “(c1, c2, c3, c4…cn),” possibly refers to counseling opportunities?

Response: We completely agree with you. It indicates counseling opportunities. We have mentioned it in the revised manuscript. (Page 5)

What was the suggested MNP intake regimen being suggested by FHW? Overall, there’s no description of how FHW were trained and how program was supposed to be implemented and be effective, other than high level saying FHW sold MNP to mothers. Reader finds out part way through the paper that FHW sell other products. Authors assume reader understands their program without providing information. More details about program design would help reader understand results (e.g., how are FHW recruited, trained, how often are they supposed to get supplies to sell, interact with caregivers? Authors reported trouble counseling mothers, but were they even trained to address this?)

Response: Thank you so much for your valuable comments. As you suggested we have incorporated program details including FHWs recruitment, training, roles and activities in the revised manuscript. (Page 7)

Switches from FHW to CHW mid-way through paper.

Response: Thank you for noting this. As you suggested we have made correction and used FHW throughout the revised manuscript.

Further details (e.g., #, %) about how high/low performers and low sales were defined and assessed, this would improve reader understanding of differences between the two groups. Also, paper explicitly says anemia reduced as a criteria for high/low but no evidence provided.

Response: Thank you for noting this. In the revised manuscript, we have given examples of criteria such as percentage points of anemia reduction for defining low and high performing districts. (Page 9)

Reports number when saturation occurred but not planned number of interviews, what were potential # to be interviewed? It’s unclear in methods total number of FGD vs individual interviews (or other types) conducted, suggest presenting in a table with numbers described for each (e.g., was it 2 FGD with 15 mothers total, for example? Noted later some of this is stated in results, but that’s too late and overall unclear to reader how many individual vs FGD vs other). Noting samples sizes are low and should be stated as a limitation, minimum sample size for stratification in qualitative data collection is 3 (and that’s low) and authors note 2 per multiple stratifications, recognizing limited number of districts. Achieving saturation at 2 interviews is a limitation.

Response: Thank you for your valuable comments. As you suggested we have given a table containing number of in-depth interviews, FGDs and key informant interviews in the revised manuscript. We have also mentioned the planned number of interviews and FGDs. (Page 10-11)

Was a software used for coding and analysis?

Response: Thanks for your comments. We used Microsoft Word Document to tabulate the textual data and manually codded on those. We have written this in the revised version of the manuscript. (Page 13)

Suggest avoiding acronyms for BRAC job descriptions - for readers unfamiliar it’s a challenge to remember what they mean and there are multiple used throughout the paper.

Response: Thanks for noting this. As you suggested we have omitted acronyms of BRAC job descriptions throughout the paper.

“On due payment” means to pay later?

Response: Thanks for your question. Yes, we have written ‘Over-due payment’ meaning pay later. In the revised manuscript, we have mentioned it in parenthesis for making it reader-friendly. (Page 20)

It seems there’s uncontrolled delivery of interventions in the area that are not coordinated or known (?) by government with so many delivering MNP for free or different price points. It would be impossible to be successful with higher cost product in this environment unless others were perceived inferior. Deserves mention of role of govt to approve interventions and coordinate etc.

Response: We agree with you. We have mentioned this in the discussion section of the revised manuscript. (Page 27)

The design was focused on high and low performers, which is a good idea, but the authors do not present/discuss results using that stratification. Were there no differences? Either way, this should be noted in results or discussion.

Response: Thanks for your valuable comments. Although the main evaluation was conducted in the low and high performing districts, in this paper we focused on overall market-based approach to promoting home fortification of diets with MNP through FHWs regardless of study areas. In addition, during data analysis we did not find any differences in the promotion of home fortification of diets with micronutrient powder through FHWs in the study areas. As you suggested we have mentioned it at the beginning of result section. (Page 14)

Reference #1 has been updated for WHO in 2020 and 2011 is old/out of date. Appreciate there has not been too much published on market-based sales of MNP, but noted many references are ~10 y old and the authors tend to cite themselves.

Response: Thank you for your comment. We have replaced the reference#1with the updated one and accordingly modified the texts.

Reviewer #2: I found this paper relevant and useful for the health sector of Bangladesh. A particular timeline of intervention and how it impacted the overall adaption of strategies into government level of sustainability could have been added. However, the paper as it is currently publishable and will benefit global research community.

Response: Thank you for your valuable comments. As you suggested we have incorporated particular timeline of intervention in the revised manuscript. We have also discussed the overall impact of it on the overall adaptation of strategies into government level of sustainability in the discussion section of the revised manuscript. (Page 27)

---

## [Decision Letter · Decision Letter 1]

7 Mar 2023

Market-based approach to promoting home fortification of diets with micronutrient powder through volunteer frontline health workers: A qualitative study

PONE-D-22-04029R1

Dear Dr. Rahman,

We’re pleased to inform you that your manuscript has been judged scientifically suitable for publication and will be formally accepted for publication once it meets all outstanding technical requirements.

Kind regards,

Ilias Mahmud, PhD

Academic Editor

PLOS ONE

Additional Editor Comments (optional):

Reviewers' comments:

Reviewer's Responses to Questions

**Comments to the Author**

1. If the authors have adequately addressed your comments raised in a previous round of review and you feel that this manuscript is now acceptable for publication, you may indicate that here to bypass the “Comments to the Author” section, enter your conflict of interest statement in the “Confidential to Editor” section, and submit your "Accept" recommendation.

Reviewer #1: All comments have been addressed

Reviewer #2: All comments have been addressed

2. Is the manuscript technically sound, and do the data support the conclusions?

Reviewer #1: Yes

Reviewer #2: Yes

3. Has the statistical analysis been performed appropriately and rigorously? 

Reviewer #1: N/A

Reviewer #2: N/A

4. Have the authors made all data underlying the findings in their manuscript fully available?

Reviewer #1: Yes

Reviewer #2: Yes

5. Is the manuscript presented in an intelligible fashion and written in standard English?

Reviewer #1: Yes

Reviewer #2: Yes

6. Review Comments to the Author

Reviewer #1: (No Response)

Reviewer #2: (No Response)

7. PLOS authors have the option to publish the peer review history of their article (what does this mean?). If published, this will include your full peer review and any attached files.

Reviewer #1: No

Reviewer #2: **Yes: **Shmim Ahmed

---

## [Editor Report · Acceptance letter]

15 Mar 2023

PONE-D-22-04029R1 

Market-based approach to promoting home fortification of diets with micronutrient powder through volunteer frontline health workers: A qualitative study 

Dear Dr. Rahman:

I'm pleased to inform you that your manuscript has been deemed suitable for publication in PLOS ONE. Congratulations! Your manuscript is now with our production department. 

Kind regards, 

on behalf of

Dr. Ilias Mahmud 

Academic Editor

PLOS ONE